# Omentin: A Key Player in Glucose Homeostasis, Atheroprotection, and Anti-Inflammatory Potential for Cardiovascular Health in Obesity and Diabetes

**DOI:** 10.3390/biomedicines12020284

**Published:** 2024-01-26

**Authors:** Cristina M. Sena

**Affiliations:** Institute of Physiology, iCBR, Faculty of Medicine, University of Coimbra, 3000-548 Coimbra, Portugal; csena@ci.uc.pt

**Keywords:** omentin, adipokines, diabetes mellitus, endothelial dysfunction, inflammation, oxidative stress

## Abstract

Omentin is an adipokine mainly produced by visceral fat tissue. It has two isoforms, omentin-1 and omentin-2. Omentin-1 is predominantly secreted by visceral adipose tissue, derived specifically from the stromal vascular fraction cells of white adipose tissue (WAT). Levels of omentin-1 are also expressed in other WAT depots, such as epicardial adipose tissue. Omentin-1 exerts several beneficial effects in glucose homeostasis in obesity and diabetes. In addition, research has suggested that omentin-1 may have atheroprotective (protective against the development of atherosclerosis) and anti-inflammatory effects, potentially contributing to cardiovascular health. This review highlights the potential therapeutic targets of omentin-1 in metabolic disorders.

## 1. Introduction

Adipose tissue, in addition to storing energy, secretes a range of cellular components and inflammatory mediators. Tumor necrosis factor-α (TNFα), interleukin-6 (IL-6), leptin, retinol-binding protein, resistin, adiponectin, omentin, apelin, visfatin, and other substances are secreted by different adipose tissue depots with regional heterogeneity [1]. These secretions have an impact on the metabolism of carbohydrates and lipids, and they also play a significant role in several pathological processes, including insulin resistance, type 2 diabetes, atherosclerosis, inflammation, and dysfunction of the vascular endothelium [1]. 

Omentin-1 (also known as intelectin-1) is a glycoprotein that has emerged as a key player in the complex interplay between adipose tissue and various physiological processes. Omentin-1 is primarily produced by the stromal vascular cells of the visceral adipose tissue [2]. This adipokine exists in human blood, and it is highly expressed in human visceral and epicardial adipose tissues, with lower levels found in other white adipose tissue (WAT) depots such as subcutaneous WAT [3]. Omentin-1 is expressed in various cells including endothelial cells, mesothelial cells, vascular cells, and intestinal Paneth cells, among others, and it exerts a paracrine, autocrine, and endocrine signaling influence [4]. Importantly, and similarly to adiponectin, circulating omentin-1 levels are reduced in obese subjects. In fact, individuals with poor glucose regulation have lower serum levels of omentin-1, and this depletion may play a role in the emergence of insulin resistance, type 2 diabetes, obesity, and metabolic syndrome. Indeed, there is a negative correlation between the serum concentration of omentin-1 and the following: body mass index, insulin resistance index, leptin, plasma glucose, fasting insulin, TNFα, and IL-6 [5,6,7,8]. This adipokine is downregulated in association with obesity-linked metabolic disorders, including type 2 diabetes and insulin resistance [5,9,10,11]. 

Several factors can influence the production and secretion of omentin-1, including obesity, insulin sensitivity, inflammation, genetic factors, and hormones, such as adiponectin and insulin. Fibroblast growth factor-21 and dexamethasone may also influence omentin-1 production [4]. Adiponectin, another adipokine, has been linked to omentin-1, and insulin sensitivity may play a role in omentin-1 regulation. Understanding these factors is essential for unraveling the complex regulatory mechanisms of omentin-1 and its potential implications for metabolic and cardiovascular health. Ongoing research continues to explore the intricate interplay between omentin-1, adipose tissue, and systemic physiology. 

This review highlights the potential therapeutic targets of omentin-1 in metabolic disorders. Relevant pre-clinical and clinical studies were summarized and discussed. A PubMed search was performed for the years between 1990–2023 using the keywords omentin-1, endothelial function, obesity, type 2 diabetes, inflammation, and oxidative stress.

## 2. Vascular Effects of Omentin-1

Omentin-1 has been implicated as having several positive effects on vascular function, making it an area of interest in cardiovascular research. Omentin-1 exerts several positive influences on vascular function, including promoting vasodilation, anti-inflammatory actions, and potential anti-atherosclerotic effects [12].

### 2.1. Vasodilation

Omentin-1 has been associated with the promotion of vasodilation, which is the relaxation of blood vessels [13]. This vasodilatory effect is important for maintaining proper blood flow and reducing resistance within the vascular system [14]. Enhanced vasodilation contributes to optimal cardiovascular function [15]. The mechanism underlying omentin-1 vasodilation involves the stimulation of nitric oxide (NO) production in endothelial cells. Nitric oxide is a signaling molecule with vasodilatory properties. Increased NO production helps regulate blood vessel tone, ensuring proper blood flow and reducing the risk of vascular constriction [15]. Omentin-1 has been shown in earlier research to protect endothelial cells by inducing NO production and endothelial NO synthase (eNOS) activation [13,16]. Omentin-1 has a cardioprotective effect as a NO-mediated vasodilator [17].

### 2.2. Anti-Inflammatory Actions

Omentin-1 exhibits anti-inflammatory properties. Inflammation plays a crucial role in the development of vascular diseases, including atherosclerosis [15]. By exerting anti-inflammatory actions, omentin-1 may help mitigate the inflammatory processes within blood vessels, reducing the risk of vascular damage and atherosclerotic plaque formation [18,19]. Indeed, omentin-1 inhibits TNF-α, IL-6, and other inflammatory cytokines, ultimately impacting vascular and tissue functions [20,21,22]. Several studies have described the anti-inflammatory and anti-atherosclerotic effects of omentin-1 through intracellular signaling pathways involving Mitogen-activated protein kinases [p38 MAPK, c-Jun N-terminal kinase (JNK), and Extracellular Signal–Regulated Kinase (ERK)], nuclear factor-κB, and AMP-activated protein kinase/protein kinase B (Akt) [4]. 

The anti-inflammatory properties of omentin-1 secreted by perivascular adipose tissue have also been considered. Through inhibiting the thioredoxin-interacting protein (TXNIP)/NLTP3 signaling pathway, omentin-1 causes an increase in the secretion of anti-inflammatory adipocytokines like IL-10 and adiponectin and a decrease in the expression of pro-inflammatory cytokines like TNF-α, IL-6, and IL-1β in obese mice [23]. Additionally, in lipopolysaccharide-induced macrophages, omentin-1 lowers oxidative stress, mitochondrial dysfunction, proinflammatory cytokines (IL-6, IL-8, and CCL2), cyclooxygenase-2, and prostaglandin E2 [22]. Omentin-1 inhibits oxidative stress in the endoplasmic reticulum to prevent vascular endothelial dysfunction. This is accomplished by stimulating the release of NO via activation of the PPAR-δ/AMP-activated protein kinase (AMPK) pathway [24]. Omentin-1 reduces the activation of NF-κB and proinflammatory agents (CCL2, IL-6, IL-1, ICAM-1, and TNF-α) in endothelial cells, which is induced by free fatty acids [25].

### 2.3. Endothelial Protection

Omentin-1 appears to have protective effects on the endothelium, the inner lining of blood vessels. Endothelial health is essential for maintaining vascular integrity and function [15,26]. In patients with type 2 diabetes, lower levels of omentin-1 have been linked to endothelial dysfunction [27]. Omentin-1 has an endothelial-dependent effect on the vascular reactivity of isolated blood vessels [13]. Accordingly, we discovered that omentin-1 treatment restored endothelial dysfunction in type 2 diabetes by normalizing ACh-induced relaxation of aortic rings in diabetic Goto-Kakizaki (GK) rats fed a high-fat diet. In arteries without perivascular adipose tissue (PVAT), omentin-1 had no effect on endothelial-independent vasorelaxation. Moreover, the aortas of diabetic GK rats mounted with PVAT showed a reduction in the endothelin 1-induced constrictor response in omentin-1 treated rats [28]. Notably, ex vivo omentin-1 vasorelaxation in aortic rings appears to be independently effective and mostly unrelated to increased peripheral insulin sensitivity [2]. Furthermore, omentin-1 was able to raise NO metabolites in the aorta and considerably raise the ratio of p-eNOS to total eNOS, suggesting that the ability of omentin-1 to restore endothelial function is facilitated by an increase in NO bioavailability [28]. Vascular oxidative stress was shown to have decreased and NO bioavailability to have improved following omentin-1 treatment in diabetic GK rats [28]. 

In addition, omentin-1 can improve endothelial function in the arteries of normal mice that have had endothelial dysfunction induced through high glucose concentrations. This improvement is mediated by AMPK and PPARδ and results in an increase in Akt/eNOS activity and NO production [24]. 

Omentin-1’s positive influence on endothelial cells contributes to the prevention of endothelial dysfunction, a key factor in the development of cardiovascular diseases [24,29,30]. Indeed, omentin-1 is negatively correlated with carotid intima-media thickness [31].

### 2.4. Anti-Atherogenic Effects

Omentin-1 has enormous potential against atherosclerotic initiation and progression. Atherosclerosis is a condition characterized by the buildup of fatty deposits (plaque) in the arterial walls, leading to narrowing and hardening of the arteries [32]. 

The ability that Omentin-1 has to promote vasodilation, reduce inflammation, and protect endothelial cells may contribute to its potential anti-atherogenic properties [28,33]. Indeed, omentin-1 has anti-inflammatory, antioxidant, and anti-apoptotic properties, positively impacting endothelial function and preventing atherosclerosis [7,33,34,35]. 

Recent research has demonstrated the extensive involvement of omentin-1 in numerous pathophysiological processes, including atherogenesis, obesity, insulin resistance, inflammatory responses, and regulation of vascular endothelial function [24,35,36,37,38]. It improves insulin sensitivity, lowers inflammation, prevents atherosclerosis, controls the activity of vascular endothelial cells [28,39], and protects the cardiovascular system [17]. 

## 3. Metabolic Regulation

Omentin-1 has gained attention due to its potential significance in vascular function and its role in metabolism. Metabolic risk factors are correlated with plasma omentin-1 levels [40]. Omentin-1 is a protein that has been studied for its potential roles in metabolic health and regulation. Omentin-1 has been implicated in metabolic regulation influencing insulin sensitivity and glucose and lipid metabolism [2,4,28,41]. However, knowledge concerning the mechanisms involved are scarce so far. Brunetti and co-workers suggested that omentin-1 could be involved in regulation of appetite [42,43]. 

### 3.1. Insulin Sensitivity

Omentin-1 has been associated with improvements in insulin sensitivity [4,28]. Insulin sensitivity refers to how effectively cells respond to insulin signaling to uptake glucose from the bloodstream. Enhanced insulin sensitivity is generally considered beneficial for metabolic health as it helps maintain normal blood glucose levels.

Omentin-1 levels have been demonstrated to be lowered in dysmetabolic conditions, including diabetes mellitus [5,11,44], obesity [11], and impaired glucose tolerance [45]. Indeed, omentin-1 is downregulated by glucose/insulin levels [4]. On the other hand, following aerobic exercise [46], hypocaloric weight loss [47], and metformin therapy [48], omentin-1 levels are increased. Higher levels of omentin-1 are advantageous for improving insulin-stimulated glucose transport because they activate Akt signaling, which regulates downstream processes like glucose metabolism [2,4]. Additionally, the researchers found that omentin-1 did not independently stimulate basal glucose transport; rather, it only improved insulin-mediated glucose transport, suggesting that it lacked intrinsic insulin-mimic activity [2]. Thus, similar to adiponectin, omentin-1 increases insulin sensitivity through an increment in insulin-mediated glucose uptake in adipose tissue. Furthermore, as a result of AMPK activation, omentin-1 has been found to inhibit the mammalian target of rapamycin (mTOR-p70S6K), which in turn enhances the activity of insulin receptor substrate [49]. 

Prior research indicates that omentin-1 plays a significant role in regulating insulin sensitivity. Chronic omentin-1 infusion into ApoE−/− mice may improve insulin resistance via PPARγ, leading to a decrease in plasma glucose concentration [2]. In addition, we have previously demonstrated that omentin-1 could lower insulin resistance in diabetic GK rats fed with a high-fat diet [28]. These beneficial metabolic effects may indirectly contribute to vascular health by addressing risk factors for cardiovascular diseases.

Chronic inflammation is associated with insulin resistance. The anti-inflammatory properties of omentin-1 may contribute to improved insulin sensitivity by reducing inflammation in adipose tissue and liver tissue, among others [6,50,51]. Inflammation interferes with insulin signaling pathways, and by mitigating inflammation, omentin-1 may help maintain proper insulin responsiveness. Omentin-1 blunts cytokine expression in different cell types [20,21,22,25,52] and is negatively associated with systemic inflammatory markers such as TNFα and IL-6 [53]. Thus, omentin-1 can be considered a biomarker for metabolic health that may function to dampen obesity-related cytokine effects [23,31]. It is possible that omentin-1 works by blocking the NF-κB pathway and triggering the AMPK- and Akt-dependent pathways [54]. Anti-diabetic medications may have an impact on the level of circulating omentin-1, which is inversely linked to the incidence of type 2 diabetes and associated complications, such as cardiomyopathy, retinopathy, and diabetic vascular disease [55]. 

Omentin-1 has also been associated with beneficial effects against bone metabolic disorders [56,57]. Understanding its role in these processes is essential for exploring its potential therapeutic applications in metabolic disorders. 

### 3.2. Glucose Uptake

Omentin-1 appears to influence glucose uptake in peripheral tissues, such as skeletal muscle and adipose tissue [2]. By promoting glucose uptake, omentin-1 may contribute to the regulation of blood glucose levels. This effect is particularly relevant in the context of insulin resistance, a condition where cells become less responsive to insulin, leading to elevated blood sugar levels [2,28,41].

### 3.3. Lipid Metabolism

Omentin-1 has been suggested to influence lipid metabolism [6,50]. It may play a role in regulating the breakdown of fats (lipolysis) and lipid storage in adipose tissue [6,50]. By modulating lipid metabolism, omentin-1 could impact insulin sensitivity and overall metabolic health.

### 3.4. Regulation of Food Intake

According to Brunetti and co-workers, omentin-1 may play a role in appetite regulation. Researchers found that omentin-1 (8 µg/kg body weight) infused centrally into Wistar rats did not cause any acute changes in food intake or the expression of genes related to agouti-related protein (AgRP), neuropeptide Y (NPY), orexin-A, amphetamine-regulated transcript (CART), corticotropin-releasing hormone (CRH), and pro-opiomelanocortin (POMC) in the hypothalamus [42]. On the other hand, omentin-1 increased food intake (starting on day 10) and body weight (starting on day 12) when it was given intraperitoneally for 14 days.

Omentin-1 was found to have a neuromodulatory function on peptidergic and aminergic signaling in both in vitro and in vivo experiments. Omentin-1 decreases the expression of the CART and CRH genes in the hypothalamus, but not that of NPY, POMC, AgRP, or orexin-A [43]. It was also noted that there were no alterations in the synthesis of dopamine or serotonin, but rather an increase in hypothalamic L-dopa and norepinephrine [43].

### 3.5. Potential Hormonal Interactions

Omentin-1’s effects on metabolic regulation may involve interactions with various hormones, including insulin, adiponectin, and others. These interactions contribute to the complex network of signaling pathways that regulate glucose and lipid metabolism [6,50].

Omentin-1 and adiponectin share structural and functional similarities. Omentin-1 may interact with adiponectin receptors, and both adipokines have been linked to improvements in insulin sensitivity [58]. The specific mechanisms through which omentin-1 and adiponectin cooperate in metabolic regulation are still an area of active research.

While the evidence suggests a link between omentin-1 and metabolic regulation, the precise mechanisms involved are still being uncovered. The potential therapeutic applications of omentin-1 in addressing metabolic disorders, such as insulin resistance and type 2 diabetes, warrant further exploration and investigation.

## 4. Clinical Implications

Some clinical implications and potential therapeutic applications of omentin-1 in cardiovascular and metabolic disorders are here summarized.

### 4.1. Cardiovascular Diseases

Omentin-1 levels are lower in coronary atherosclerosis patients than in healthy individuals [31]. In addition, omentin-1 levels are also lower in the coronary endothelium in patients with coronary artery disease (CAD) [33], a finding that may be related to a negative feedback regulation mechanism. On the other hand, Saely et al. [59] discovered that elevated plasma omentin-1 was a predictor of cardiovascular events in patients with CAD after analyzing plasma omentin-1 in patients undergoing coronary angiography [59]. Potential racial differences may be the cause of these contradictory findings, underscoring the need for additional research and analysis. Omentin-1 is elevated in middle layer vascular smooth muscle cells, circulation, and macrophage-induced foam cells in coronary plaques in patients with acute coronary syndrome [33]. High counteracting atherosclerosis reactivity may be linked to elevated levels of omentin-1 in patients with acute coronary syndromes or diabetes [33,60]. Moreover, in patients with type 2 diabetes, even after accounting for other cardiovascular risk factors and a thorough medication history, there was still an independent correlation found between circulating omentin-1 level and carotid plaque and arterial stiffness [61]. Accordingly, patients with diabetic ulcers showed lower mean diastolic blood pressure values, lower serum levels of omentin-1, lower endothelial function values and higher body weight values than healthy controls [62]. Additionally, research has revealed that in hemodialysis patients with suspected CAD, heart failure, or subclinical atherosclerosis, serum omentin-1 levels are a significant predictor of cardiovascular events [59,63,64]. Thus, disease progression and co-mobilities may also be an important factor to understand the role of omentin-1 in cardiovascular disease.

Omentin-1 secreted from epicardial adipose tissue (EAT) exhibits cardioprotective properties. Previous studies have suggested that omentin-1 levels have a cardioprotective role and are linked with cardiovascular function [65]. The authors studied the function of omentin-1 in atrial fibrosis, which can result in heart failure. Remarkably, by blocking the transforming growth factor-β signaling pathways, omentin-1 acts as an anti-fibrotic adipokine that reverses the effects of atrial fibrosis [65]. A negative correlation between the production of adipocytokines from EAT and coronary atherosclerosis was observed by Verhagen and co-workers [66]. Omentin-1 mRNA levels were found to be significantly lower in stenotic coronary artery segments surrounding EAT. As a result, there may be a negative correlation between coronary atherosclerosis and omentin-1 mRNA expression in EAT [35]. It is important to highlight the paracrine effects of EAT because the association between EAT-derived adipocytokines and coronary atherosclerosis has been underappreciated. In actuality, the delicate balance between adipocytokines that promote and inhibit inflammation is prone to breakdown under pathological circumstances [67]. Verhagen and collaborators demonstrated that, in contrast to non-stenotic segments, adiponectin mRNA expression in EAT near stenotic segments was not as low as predicted. Additionally, there was a noticeable decrease in pro-inflammatory adipocytokine secretion from EAT adjacentnt to the stenotic coronary segments [66]. Adipocytokines from EAT have a pro-inflammatory proactive profile that is generally more pronounced in patients with CAD than in healthy individuals [68]. In patients with CAD, the regulation of adipocytokines from EAT near stenotic coronary segments seems to be complex. 

According to Zhu and collaborators [69], there was a significant correlation between omentin-1 levels in acute myocardial infarction and post-infarction myocardial function. The findings showed that omentin-1 inhibits negative remodeling and also clarified the connection between omentin-1 and myocardial ischemia or reperfusion. Future research will resolve any remaining doubts regarding the matter. Serum omentin-1 concentrations were found to be inversely correlated with the development of atrial fibrillation and atrial remodeling in a study by Tao et al. [70]. Reduced serum omentin-1 levels were shown by Onur and co-workers to be an independent predictor of CAD and to be linked to the severity of disease in women with postmenopausal CAD [71]. Serum omentin-1 levels were shown by Narumi et al. to be predictive of cardiovascular events in heart failure patients. In this population, serum omentin-1 levels seem to be a novel risk classification prognostic factor [63].

Omentin-1’s positive effects on vascular function, including vasodilation, anti-inflammatory actions, and potential anti-atherosclerotic effects, suggest potential applications in cardiovascular diseases [72,73]. Therapies aimed at increasing omentin-1 levels or enhancing its activity could be explored for conditions such as atherosclerosis, hypertension, and other cardiovascular disorders [45,55,74].

### 4.2. Metabolic Disorders

Omentin-1’s involvement in metabolic regulation, insulin sensitivity, and glucose metabolism make it a potential target for treating metabolic disorders. Strategies to modulate omentin-1 levels or activity may be considered in the management of insulin resistance, type 2 diabetes [73], and obesity [23,75].

#### 4.2.1. Obesity and Metabolic Syndrome 

In obesity associated with insulin resistance, higher circulating levels of retinol-binding protein 4 [76], visfatin [77], chemerin [75], vaspin [78], and resistin [79] and lower levels of omentin-1 [63] and adiponectin [80] have been reported.

De Souza Batista et al. [9] assessed omentin-1 concentrations in obese individuals. Compared to overweight and obese patients, those with normal body weights had higher levels of omentin-1. Studies on children with obesity [81] and women with obesity [82] produced similar findings. Additionally, omentin-1 concentrations in men with normal body mass were significantly higher than those in men who were overweight or obese, according to Moreno-Navarette et al. [82]. In contrast, Rahimlou and co-workers, when studying obese and normal weight individuals [83], did not obtain similar findings. Orlik et al. also presented contradicting findings [84]. It is important to remember that a low-fat, hypocaloric diet can cause weight loss, which can cause serum omentin-1 levels to rise significantly. 

Omentin-1’s role in adipose tissue and its potential influence on lipid metabolism may have implications for obesity and metabolic syndrome [7,38,45]. Therapeutic approaches that aim to modulate omentin-1 levels or activity could be explored in the context of obesity management and preventing metabolic syndrome-related complications. Indeed, omentin-1 can control insulin sensitivity and glucose metabolism because it circulates, which may stop the advancement of CAD in obese patients. Waist circumference and HOMA-IR have a negative correlation with serum omentin-1. Increased diastolic cardiac function following pioglitazone treatment is correlated with increased serum omentin-1 levels in the majority of obese patients with diabetes and low cardiovascular risk who underwent bariatric surgery [17]. 

Margaritis and co-workers [85] reported that adipokine expression levels in some adipose tissue depots do not always correlate with adipokine levels in circulation, suggesting intricate mechanisms governing the biology and secretome of adipose tissue [85]. Importantly, in obesity, omentin-1 plays a significant anti-inflammatory role, most likely through upregulating Th-2 cytokines like IL-13 and IL-14. Increased concentrations of omentin-1 are thought to lower the levels of inflammatory cytokines [53].

Furthermore, the longitudinal relationship between omentin-1 concentrations and the risk of type 2 diabetes was assessed by Wittenbecher et al. (2016). 2500 randomly chosen members of the Potsdam cohort of the European Prospective Investigation into Cancer and Nutrition (EPIC) served as the basis for this observational study. In prospective analyses, the authors found no evidence of omentin-1’s protective effect against diabetes, despite omentin-1’s direct associations with adiponectin and inverse associations with measurements of body fat. Surprisingly, data revealed that individuals with high adiponectin concentrations had an increased risk of diabetes related to omentin-1 [86]. 

In type 1 diabetes, Polkowska and co-workers (2016) noted that, regardless of the length of this pathology, children with this type of diabetes had serum omentin-1 concentrations that were significantly lower than those of control children [87]. 

#### 4.2.2. Insulin Resistance and Type 2 Diabetes

Improving insulin sensitivity is a key goal in managing insulin resistance and type 2 diabetes. Omentin-1’s potential to enhance insulin sensitivity suggests that it could be a therapeutic target for individuals with insulin resistance or those at risk of developing type 2 diabetes [23,44,53]. Insulin resistance and type 2 diabetes are inversely correlated with serum omentin-1 levels [10]. 

### 4.3. Inflammatory Disorders

In recent years, the multifaceted properties of omentin-1 have garnered increased attention, particularly considering the potential impact that it exhibits on inflammation and its subsequent influence on various physiological processes. Omentin-1’s anti-inflammatory properties make it relevant in conditions associated with chronic inflammation. This includes inflammatory disorders that can impact vascular health, such as rheumatoid arthritis [88]. Omentin-1-based interventions might be investigated as a complementary approach to address inflammation in these conditions.

This adipokine can suppress inflammation in adipose tissue. In one study, omentin-1 treatment decreased pro-inflammatory cytokines, reversed macrophage polarization, and reduced inflammation in mice given a high-fat diet [23]. Omentin-1 also reduces the transcription factor NF-kB by blocking the Akt pathway in the inflammation signaling, and the secretion of TNF-α and other pro-inflammatory cytokines are decreased when NF-kB is inhibited [48]. Therefore, omentin-1 has the potential to be used as a therapy to reverse pathological conditions and significantly reduce inflammation in patients with conditions such as steatotic liver disease or osteoarthritis [89,90].

### 4.4. Future Therapeutic Developments

Ongoing research may uncover novel therapeutic strategies, such as the development of omentin-1-based drugs or interventions that target omentin-1 receptors. These advancements could open new avenues for personalized medicine approaches tailored to individuals with specific vascular and metabolic profiles.

Importantly, while the potential therapeutic applications of omentin-1 are promising, further research, including clinical trials, is needed to establish its efficacy, safety, and optimal modes of administration. The field of omentin-1 research is dynamic, and advancements in understanding its role in health and disease may lead to new therapeutic opportunities in the future.

## 5. Challenges and Future Directions

While omentin-1 holds promise for its potential benefits in vascular function, there are several challenges and gaps in knowledge. The precise molecular mechanisms through which omentin-1 exerts its effects on vascular cells are not fully understood [28]. It is necessary to identify the specific receptors and downstream signaling pathways involved in omentin-1-mediated vasodilation, anti-inflammatory action, and other vascular effects. 

In addition, omentin-1 is expressed in various tissues, including adipose tissue, but its effects may be tissue specific. Understanding omentin-1 functions in different tissues and whether its effects on vascular function vary in different vascular beds is a complex aspect that requires further investigation.

Omentin-1 shares similarities with other adipokines, such as adiponectin or apelin. The interactions and potential synergies between omentin-1 and other adipokines in modulating vascular function are not fully understood. Disentangling these interactions is crucial for a more comprehensive view of omentin-1’s role.

### 5.1. Biomarker Validity and Limited Clinical Data 

Omentin-1 has been proposed as a potential biomarker for certain metabolic and cardiovascular conditions [59]. However, the validity, specificity, and sensitivity of omentin-1 as a biomarker need to be thoroughly validated in diverse populations and clinical settings. 

Additionally, while preclinical studies suggest beneficial effects, the number of clinical studies investigating omentin-1’s role in vascular function is relatively limited. Future research on omentin-1 could explore several key areas to deepen our understanding of its role in vascular function and facilitate the development of targeted therapeutic interventions. Expanding the clinical evidence base is critical for understanding its relevance in human health and disease.

### 5.2. Genetic Variations

Previous studies described that compared to people with the wild genotype, carriers of the adipose tissue genotype rs2274907 SNP in the omentin-1 gene may be four times more likely to develop CAD without changes in serum lipid levels [91]. The expression of the AT genotype of rs2274907A>T also increased sensitivity to CAD, as demonstrated by Jha et al. [92]. Additionally, these researchers discovered a connection between the genotype distribution of rs2274907A>T SNP and the levels of HDL, LDL, and total cholesterol. These findings align with related research in the literature [93,94]. 

The impact of genetic variations in the omentin-1 gene on omentin-1 expression and function needs further elucidation [95]. In addition, it is important to investigate whether specific genetic polymorphisms are associated with altered susceptibility to vascular and metabolic disorders [96].

## 6. Conclusions

A schematic diagram summarizing the key actions of omentin-1 is presented (Figure 1).

Omentin-1 is an adipocytokine with antioxidative and anti-inflammatory properties that is widely expressed in a variety of cells. This adipokine exhibits a wide range of therapeutic potentials in diabetes treatment through its actions in reducing comorbidities linked to type 2 diabetes mellitus, such as vascular diseases and diabetic nephropathy [97,98]. In addition, omentin-1 inhibits insulin resistance, atherosclerosis, and inflammation through the intracellular signaling pathways of AMP-activated protein kinase/Akt/nuclear factor-κB/mitogen-activated protein kinase (ERK, p38, and JNK). Notably, of the major adipocytokines that inhibit atherosclerosis, omentin-1 has a strong correlation with inflammation, macrophage differentiation, arterial calcification, and plaque formation [99]. Omentin-1 may be used as a biomarker for metabolic syndrome, obesity, diabetes, atherosclerosis, ischemic heart disease, and inflammatory diseases. This review sheds light on how omentin-1 might be used to treat these diseases and serve as a biomarker.

## Figures and Tables

**Figure 1 biomedicines-12-00284-f001:**
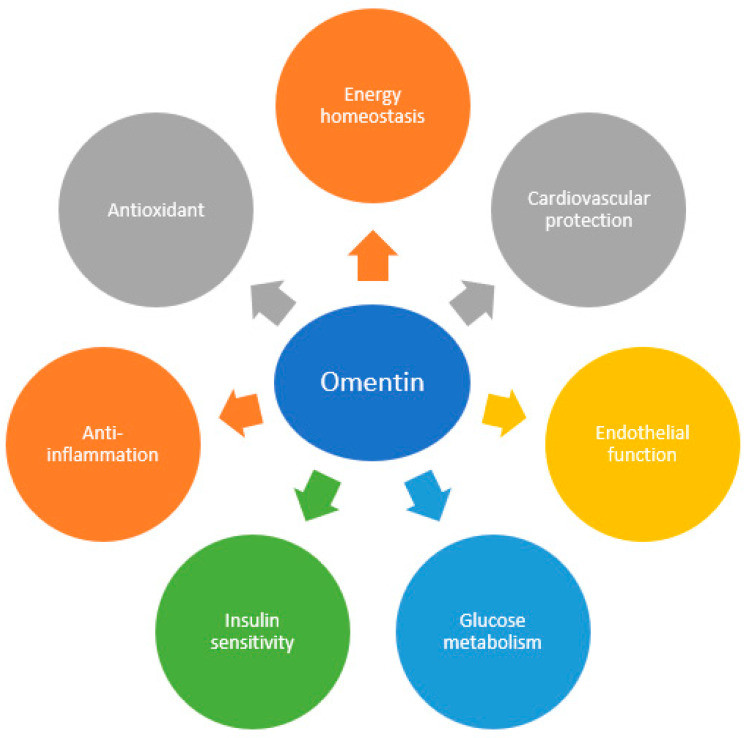
Schematic representation of the beneficial effects of omentin-1.

## Data Availability

Not applicable.

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
