# Peer review of "Omentin: A Key Player in Glucose Homeostasis, Atheroprotection, and Anti-Inflammatory Potential for Cardiovascular Health in Obesity and Diabetes"

_biomedicines, 2024, doi:10.3390/biomedicines12020284_

Round 1

Reviewer 1 Report

Comments and Suggestions for Authors

The article presents a topic of great interest and it is well written 

I suggest author to add

-          Omentin and arterial stiffness

https://journals.lww.com/jhypertension/abstract/2021/04001/the_association_between_arterial_stiffness_and.605.aspx

-          Omentin and cardiovascular risk

https://link.springer.com/article/10.1007/s00125-019-05017-2

Also some sentences must be rephrased

Explore the impact of genetic variations in the omentin gene on omentin expression and function (Rathwa et al., 2019). Investigate whether specific genetic polymorphisms are associated with altered susceptibility to vascular and metabolic disorders.

Evaluate the feasibility and efficacy of developing therapeutic interventions that directly target omentin or its receptors. Investigate the potential use of omentin-based drugs or interventions to improve vascular function and metabolic health.

Elucidate the precise molecular mechanisms through which omentin exerts its effects on vascular cells. Identify the specific receptors and downstream signaling pathways involved in omentin-mediated vasodilation, anti-inflammatory actions, and other vascular effects.

ETC.

Comments on the Quality of English Language

Minor mistakes 

Author Response

Reviewer #1

The article presents a topic of great interest and it is well written 

I suggest author to add

-          Omentin and arterial stiffness

https://journals.lww.com/jhypertension/abstract/2021/04001/the_association_between_arterial_stiffness_and.605.aspx

-          Omentin and cardiovascular risk

https://link.springer.com/article/10.1007/s00125-019-05017-2

[57] Niersmann, C., Carstensen-Kirberg M, Maalmi H, Holleczek B, Roden M, Brenner H, Herder C, Schöttker B. Higher circulating omentin is associated with increased risk of primary cardiovascular events in individuals with diabetes. Diabetologia 2020,63,410–418. doi.org/10.1007/s00125-019-05017-2

These topics and references were added as follows (section 4.1):

4.1. Cardiovascular Diseases

Omentin, secreted from epicardial adipose tissue, exhibits cardioprotective properties. Previous studies have suggested omentin-1 levels have a cardioprotective role and are linked with cardiovascular function [55]. The authors studied the function of omentin-1 in atrial fibrosis, which can result in heart failure. Remarkably, by blocking the transforming growth factor-β signaling pathways, omentin-1 is an anti-fibrotic adipokine that reverses the effects of atrial fibrosis [55].

Omentin levels in the coronary endothelium are lower in patients with coronary artery disease [29], a finding that may be related to a negative feedback regulation mechanism. On the other hand, Saely et al. [56] discovered that elevated plasma omentin was a predictor of cardiovascular events in patients with coronary artery disease after analyzing plasma omentin in patients undergoing coronary angiography [56]. Potential racial differences may be the cause of these contradictory findings underscoring the need for additional research and analysis. High counteracting atherosclerosis reactivity may be linked to elevated levels of omentin-1 in patients with acute coronary syndromes or diabetes [29, 57]. Moreover, in patients with type 2 diabetes, even after accounting for other cardiovascular risk factors and a thorough medication history, there was still an independent correlation found between circulating omentin-1 level and carotid plaque and arterial stiffness [58]. Accordingly, patients with diabetic ulcers showed lower mean diastolic blood pressure values than healthy controls, lower serum levels of omentin-1, lower endothelial function values and, higher body weight values [59]. Thus, disease progression and co-mobilities may also be an important factor to understand the role of omentin-1 in cardiovascular disease.

Omentin's positive effects on vascular function, including vasodilation, anti-inflammatory actions, and potential anti-atherosclerotic effects, suggest potential applications in cardiovascular diseases [60,61]. Therapies aimed at increasing omentin levels or enhancing its activity could be explored for conditions such as atherosclerosis, hypertension, and other cardiovascular disorders [39,53, 62].

  1. Also some sentences must be rephrased

Please consider that sentences were rephrased or removed (section 5.1, 5.2, 5.3, 5.4, 5.7, 5.8, 5.9, 5.10, 5.11.

The impact of genetic variations in the omentin gene on omentin expression and function needs further elucidation [75]. In addition, it is important to investigate whether specific genetic polymorphisms are associated with altered susceptibility to vascular and metabolic disorders [76].

The precise molecular mechanisms through which omentin exerts its effects on vascular cells are not fully understood [23]. It is necessary to identify the specific receptors and downstream signaling pathways involved in omentin-mediated vasodilation, anti-inflammatory actions, and other vascular effects.

In addition, omentin is expressed in various tissues, including adipose tissue, but its effects may be tissue specific. Understanding omentin functions in different tissues and whether its effects on vascular function vary in different vascular beds is a complex aspect that requires further investigation.

ETC.

Comments on the Quality of English Language

Minor mistakes 

Thank you very much for the valuable comments. The manuscript was carefully revised. They were corrected.

Reviewer 2 Report

Comments and Suggestions for Authors

Omentin: an atheroprotective adipokine for vascular health

This review deals with the role of Omentin,  an adipokine mainly produced by visceral fat tissue, in glucose homeostasis in obesity and diabetes. In addition, research has suggested that omentin may have atheroprotective (protective against the development of atherosclerosis) and anti-inflammatory effects, potentially contributing to cardiovascular health. This review aims to highlight the potential therapeutic targets of omentin-1 in metabolic-related disorders.

Although the theme is of interest, I think this review cannot be published in the present form, and should undergo major revisions.  In addition, The English used sound readable in the first part of the paper, then it was hard to comprehend (construction of the sentences too similar to a list).

Comments:

Para 3: (metabolism?) regulation

3.2. “Glucose Uptake Omentin appears to influence glucose uptake in peripheral tissues, such as skeletal muscle and adipose tissue” Please add some references.

I would not include paragraph 3.3 (interaction with ADPN) within this section. Even if it deals with insulin sensitivity the title does not seem appropriate. This paragraph can be included in the former 3.1. Alternatively, it can be included in 3.6

Para 3.4…..Thus, omentin is a biomarker for metabolic …This sentence sounds too strong : I would suggest to be more cautious and write “omentin can be considered” or at least  “omentin can act as a biomarker”

Para 3.6 can be slightly extended.

Para 4:

Please Rephrase” Herin are summarize some clinical implications…”: into : “ herein we summarize some …” or “some clinical implications are here summarized…”

The majority of the paragraphs and subparagraph are too short and concise, consisting just of few lines. In my opinion, the Author should further extend at least some of them, because they risk appearing like a mere list of omentin ‘s action or potential clinical applications

The titles of these sub paragraphs (4.2.1; 4.2.2; 4.2.3) do not sound convincing to me, and should be revised. “Obesity” appears in 4.2.1 and 4.2.3. I can’t comprehend the rationale of splitting in two different paragraph.

In the section of inflammatory disorders (4.3) The Author can certainly expand e question. There are several updated papers in 2023 (indexed in pubmed for instance) that deal with omentin in this condition

Para 5

Starting from this paragraph, the citations are very few. Even if the para 5. Challenges and Future Directions, deals with something that may happen and thus is not reported yet, it is quite strange to see a complete lack of reference in this part of the paper (just 1 citation in paragraph5.9) especially because this paper is a review.

Is “Receptor Identification and Signaling Pathways” a title? If it is, It should be recognizable (italics? Number ?). Are there papers to be cited in this regard? I think this paragraph is too short, also considering the importance of the knowledge of the receptors for possible therapy and drug design.

5.3. Dose-Response Relationships : are there studies that report some results about this item in particular condition/model/ tissue to serve as background? IF so, Authors should include them in this paragraph to extend it, because it is pretty short  

5.5. “Biomarker Validity Omentin has been proposed as a potential biomarker for certain metabolic and cardiovascular conditions” Author should cite some references .

“Initiate longitudinal studies to investigate the association between circulating omentin levels and the development of vascular and metabolic disorders over time. This could provide insights into the predictive value of omentin as a biomarker”. This sentence is not clear to me.

5.7.” Mechanistic Insights Elucidate the precise molecular mechanisms through which omentin exerts its effects on vascular cells. Identify the specific receptors and downstream signaling pathways involved in omentin-mediated vasodilation, anti-inflammatory actions, and other vascular effects.” This sentence is not clear and the paragraph is too short to stand alone: Consider joining this content into other existing paragraphs

5.8. “Role in Inflammation Further investigate omentin's role in modulating inflammation, both locally within adipose tissue and systemically. Explore how omentin's anti-inflammatory actions may impact the progression of inflammatory diseases, including those affecting the vasculature”. This sentence is not clear and the paragraph is too short to stand alone

Para 5 sounds like a bullet point list, but without bullet points. I can’t comprehend the construction of the sentences.

After the reading of this review I have some concerns about the TITLE of the paper-“Omentin: an atheroprotective adipokine for vascular health-  I suggest to modify this title since vascular health is not the only theme presented here, but many beneficial and metabolic effects are treated. As stated in the abstract, this review deals with the role of Omentin,  in glucose homeostasis in obesity and diabetes. In addition, research has suggested that omentin may have atheroprotective (protective against the development of atherosclerosis) and anti-inflammatory effects, potentially contributing to cardiovascular health.

Author Response

Reviewer #2

Omentin: an atheroprotective adipokine for vascular health

This review deals with the role of Omentin,  an adipokine mainly produced by visceral fat tissue, in glucose homeostasis in obesity and diabetes. In addition, research has suggested that omentin may have atheroprotective (protective against the development of atherosclerosis) and anti-inflammatory effects, potentially contributing to cardiovascular health. This review aims to highlight the potential therapeutic targets of omentin-1 in metabolic-related disorders.

Although the theme is of interest, I think this review cannot be published in the present form, and should undergo major revisions.  In addition, The English used sound readable in the first part of the paper, then it was hard to comprehend (construction of the sentences too similar to a list).

Comments:

Para 3: (metabolism?) regulation

3.2. “Glucose Uptake Omentin appears to influence glucose uptake in peripheral tissues, such as skeletal muscle and adipose tissue” Please add some references.

Thank you very much for the valuable comments.

Please consider that references were added as follows:

Glucose Uptake

Omentin appears to influence glucose uptake in peripheral tissues, such as skeletal muscle and adipose tissue [2]. By promoting glucose uptake, omentin may contribute to the regulation of blood glucose levels. This effect is particularly relevant in the context of insulin resistance, a condition where cells become less responsive to insulin's actions, leading to elevated blood sugar levels [2,23,37].

I would not include paragraph 3.3 (interaction with ADPN) within this section. Even if it deals with insulin sensitivity the title does not seem appropriate. This paragraph can be included in the former 3.1. Alternatively, it can be included in 3.6

Section 3.3 was moved to section 3.6 as suggested.

Para 3.4…..Thus, omentin is a biomarker for metabolic …This sentence sounds too strong : I would suggest to be more cautious and write “omentin can be considered” or at least  “omentin can act as a biomarker”

It was changed as suggested: “omentin can be considered…”

Para 3.6 can be slightly extended.

It was extended as suggested.

Para 4:

Please Rephrase” Herin are summarize some clinical implications…”: into : “ herein we summarize some …” or “some clinical implications are here summarized…”

It was rephrased as follows:

Clinical Implications

Some clinical implications and potential therapeutic applications of omentin are here summarized in cardiovascular and metabolic disorders.

The majority of the paragraphs and subparagraph are too short and concise, consisting just of few lines. In my opinion, the Author should further extend at least some of them, because they risk appearing like a mere list of omentin ‘s action or potential clinical applications

The titles of these sub paragraphs (4.2.1; 4.2.2; 4.2.3) do not sound convincing to me, and should be revised. “Obesity” appears in 4.2.1 and 4.2.3. I can’t comprehend the rationale of splitting in two different paragraph.

Please consider that the sections were combined (4.2.1 and 4.2.3).

In the section of inflammatory disorders (4.3) The Author can certainly expand e question. There are several updated papers in 2023 (indexed in pubmed for instance) that deal with omentin in this condition

It was extended as follows:

4.3. Inflammatory Disorders

In recent years, the multifaceted properties of omentin have garnered increased attention, particularly for its potential impact on inflammation and its subsequent influence on various physiological processes. Omentin's anti-inflammatory properties make it relevant in conditions associated with chronic inflammation. This includes inflammatory disorders that can impact vascular health, such as rheumatoid arthritis [71]. Omentin-based interventions might be investigated as a complementary approach to address inflammation in these conditions.

This adipokine can suppress inflammation in adipose tissue. Indeed, omentin treatment decreased pro-inflammatory cytokines, reversed macrophage polarization, and reduced inflammation in mice given a high-fat diet [51]. Omentin reduced the transcription factor NF-kB by blocking the Akt pathway in the inflammation signaling. TNF-α secretion and other pro-inflammatory cytokines are decreased when NF-kB is inhibited [42]. Therefore, omentin has the potential to be used as a therapy to reverse pathological conditions and significantly reduce inflammation in patients with conditions such as steatotic liver disease or osteoarthritis [72,73].

Para 5

Starting from this paragraph, the citations are very few. Even if the para 5. Challenges and Future Directions, deals with something that may happen and thus is not reported yet, it is quite strange to see a complete lack of reference in this part of the paper (just 1 citation in paragraph5.9) especially because this paper is a review.

Is “Receptor Identification and Signaling Pathways” a title? If it is, It should be recognizable (italics? Number ?). Are there papers to be cited in this regard? I think this paragraph is too short, also considering the importance of the knowledge of the receptors for possible therapy and drug design.

5.3. Dose-Response Relationships : are there studies that report some results about this item in particular condition/model/ tissue to serve as background? IF so, Authors should include them in this paragraph to extend it, because it is pretty short 

5.5. “Biomarker Validity Omentin has been proposed as a potential biomarker for certain metabolic and cardiovascular conditions” Author should cite some references .

“Initiate longitudinal studies to investigate the association between circulating omentin levels and the development of vascular and metabolic disorders over time. This could provide insights into the predictive value of omentin as a biomarker”. This sentence is not clear to me.

5.7.” Mechanistic Insights Elucidate the precise molecular mechanisms through which omentin exerts its effects on vascular cells. Identify the specific receptors and downstream signaling pathways involved in omentin-mediated vasodilation, anti-inflammatory actions, and other vascular effects.” This sentence is not clear and the paragraph is too short to stand alone: Consider joining this content into other existing paragraphs

5.8. “Role in Inflammation Further investigate omentin's role in modulating inflammation, both locally within adipose tissue and systemically. Explore how omentin's anti-inflammatory actions may impact the progression of inflammatory diseases, including those affecting the vasculature”. This sentence is not clear and the paragraph is too short to stand alone

Para 5 sounds like a bullet point list, but without bullet points. I can’t comprehend the construction of the sentences.

Please consider that most sections were removed. It was changed as follows:

  1. Challenges and Future Directions

While omentin holds promise for its potential benefits in vascular function, there are several challenges and gaps in knowledge. The precise molecular mechanisms through which omentin exerts its effects on vascular cells are not fully understood [23]. It is necessary to identify the specific receptors and downstream signaling pathways involved in omentin-mediated vasodilation, anti-inflammatory actions, and other vascular effects.

In addition, omentin is expressed in various tissues, including adipose tissue, but its effects may be tissue specific. Understanding omentin functions in different tissues and whether its effects on vascular function vary in different vascular beds is a complex aspect that requires further investigation.

Omentin shares similarities with other adipokines, such as adiponectin or apelin. The interactions and potential synergies between omentin and other adipokines in modulating vascular function are not fully understood. Disentangling these interactions is crucial for a more comprehensive view of omentin's role.

5.1. Biomarker Validity and Limited Clinical Data

Omentin has been proposed as a potential biomarker for certain metabolic and cardiovascular conditions [56]. However, the validity, specificity, and sensitivity of omentin as a biomarker need to be thoroughly validated in diverse populations and clinical settings.

Additionally, while preclinical studies suggest beneficial effects, the number of clinical studies investigating omentin's role in vascular function is relatively limited. Future research on omentin could explore several key areas to deepen our understanding of its role in vascular function and facilitate the development of targeted therapeutic interventions. Expanding the clinical evidence base is critical for understanding its relevance in human health and disease.

5.2. Genetic Variations

Previous studies described that compared to people with the wild genotype, carriers of the adipose tissue genotype of the rs2274907 SNP in the omentin-1 gene may be four times more likely to develop coronary artery disease without changes in serum lipid levels [74].

The impact of genetic variations in the omentin gene on omentin expression and function needs further elucidation [75]. In addition, it is important to investigate whether specific genetic polymorphisms are associated with altered susceptibility to vascular and metabolic disorders [76].

After the reading of this review I have some concerns about the TITLE of the paper-“Omentin: an atheroprotective adipokine for vascular health-  I suggest to modify this title since vascular health is not the only theme presented here, but many beneficial and metabolic effects are treated. As stated in the abstract, this review deals with the role of Omentin,  in glucose homeostasis in obesity and diabetes. In addition, research has suggested that omentin may have atheroprotective (protective against the development of atherosclerosis) and anti-inflammatory effects, potentially contributing to cardiovascular health.

Please consider that the title was changed as suggested:

“Omentin: a key player in glucose homeostasis, atheroprotection, and anti-inflammatory potential for cardiovascular health in obesity and diabetes "

Round 2

Reviewer 2 Report

Comments and Suggestions for Authors

Dear Author.

Although most criticisms have been solved, while reading the revised version of the paper I found some other major concerns  to be addressed before this review could be published.

Major

1)     Para 2.4. Nitric Oxide Production: I think that the content of this paragraph should be splitted, because as the Author claims the NO production is at the base of two Omentin’s functions described in section 2. In my opinion the first part of para 2.4 (Omentin has been reported to stimulate the production of NO in endothelial cells. Nitric oxide is a signaling molecule with vasodilatory properties. Increased NO production helps regulate blood vessel tone, ensuring proper blood flow and reducing the risk of vascular constriction [15]. Omentin-1 has been shown in earlier research to protect endothelial cells by inducing NO production and endothelial NO synthase (eNOS) activation [13, 27]) would fit in the paragraph of vasodilation (2.1) and the last part (Furthermore, omentin-1 was able to raise NO metabolites in the aortas and considerably raise the ratio of p-eNOS to total eNOS, suggesting that omentin-1's ability to restore endothelial function is caused by an increase in NO bioavailability [23]. Vascular oxidative stress was decreased, and nitric oxide (NO) bioavailability was improved by omentin-1 treatment in diabetic GK rats [23]. Indeed, omentin-1 can improve endothelial function in normal mice's arteries that have endothelial dysfunction brought on by high glucose concentrations. This improvement is mediated by AMPK and PPARδ and results in an increase in Akt/eNOS activity and NO production [25]) in the paragraph 2.3 endothelial protection. The sentences should be somehow rewritten, serving as explanatory for the possible mechanisms underlying such effects of Omentin.

2) Para 3.3 Since this paragraph dealing with inflammation is not strictly related to metabolic effect per se’, I propose to join it within paragraph 3.1.

3) 6. Conclusion: “ A schematic diagram summarizing the key features of omentin is presented”. Maybe “effects”, “role” or “actions” should replace the word “features”

4) Scheme fig 1: “- Omentin-1 is an adipocytokine widely expressed in a variety of cells, exhibiting microbial defense, and anti-apoptotic properties”. These statement are NOT indicated in figure nor in the main text with dedicated paragraphs. I suggest to omit these actions or discuss them also in the paper (or at least add proper references)

Minor:

1)Para 3.5 “Omentin and adiponectin share structural and functional similarities. Omentin may interact with adiponectin receptors and both adipokines have been linked to improvements in insulin sensitivity [54]. The specific mechanisms through which omentin and adiponectin cooperate in metabolic regulation are still an area of active research” and 4.2.1:” Margaritis and co-workers [70] reported that adipokine expression levels in some adipose tissue depots do not always correlate with adipokine levels in the circulation suggesting intricate mechanisms governing the biology and secretome of adipose tissue [70].” Why are these sentences in Red? Do they represent the only changes made respect to the former original version of the MS? I am not sure. Maybe they should be in black.

2) If Fig 1 is taken from a published paper, please add reference in its caption

3) Finally, the way the Conclusions are presented (“- Omentin-1 is an adipocytokine widely expressed in a variety of cells, exhibiting microbial defense, antioxidative, anti-inflammatory, and anti-apoptotic properties.

 - Omentin-1 exhibits a wide range of therapeutic potential in diabetes by reducing comorbidities linked to type 2 diabetes mellitus, such as vascular diseases and diabetic nephropathy [77,78]. Noteworthy, of the major Omentin Energy homeostasis Cardiovascular protection Endothelial function Glucose metabolism Insulin sensitivity Antiinflammation Antioxidant Biomedicines 2023, 11, x FOR PEER REVIEW 8 of 12 adipocytokines that inhibit atherosclerosis, omentin-1 has a strong correlation with inflammation, macrophage differentiation, arterial calcification, and plaque formation [79].

- Omentin-1 inhibits insulin resistance, atherosclerosis, and inflammation through the intracellular signaling pathways of AMP-activated protein kinase/Akt/nuclear factor-κB/mitogen-activated protein kinase (ERK, p38, and JNK).

- Omentin-1 may be used as a biomarker for metabolic syndrome, obesity, diabetes, atherosclerosis, ischemic heart disease, and inflammatory diseases.

- This review sheds light on how omentin-1 might be used to treat these diseases and serve as a biomarker

“) is quite strange to me since they seem a list with each item preceded by a hyphen. Usually, the bullet point is used in a Highlights section. Initially I was thinking this was the caption of Scheme 1. Please, consider revising the presented text to make its role (Caption, highlights or conclusions) clearer.

 The topic is interesting but it still needs some corrections.

Author Response

Dear reviewer,

Thank you very much for your comments.

Although most criticisms have been solved, while reading the revised version of the paper I found some other major concerns  to be addressed before this review could be published.

Major

1)     Para 2.4. Nitric Oxide Production: I think that the content of this paragraph should be splitted, because as the Author claims the NO production is at the base of two Omentin’s functions described in section 2. In my opinion the first part of para 2.4 (Omentin has been reported to stimulate the production of NO in endothelial cells. Nitric oxide is a signaling molecule with vasodilatory properties. Increased NO production helps regulate blood vessel tone, ensuring proper blood flow and reducing the risk of vascular constriction [15]. Omentin-1 has been shown in earlier research to protect endothelial cells by inducing NO production and endothelial NO synthase (eNOS) activation [13, 27]) would fit in the paragraph of vasodilation (2.1) and the last part (Furthermore, omentin-1 was able to raise NO metabolites in the aortas and considerably raise the ratio of p-eNOS to total eNOS, suggesting that omentin-1's ability to restore endothelial function is caused by an increase in NO bioavailability [23]. Vascular oxidative stress was decreased, and nitric oxide (NO) bioavailability was improved by omentin-1 treatment in diabetic GK rats [23]. Indeed, omentin-1 can improve endothelial function in normal mice's arteries that have endothelial dysfunction brought on by high glucose concentrations. This improvement is mediated by AMPK and PPARδ and results in an increase in Akt/eNOS activity and NO production [25]) in the paragraph 2.3 endothelial protection. The sentences should be somehow rewritten, serving as explanatory for the possible mechanisms underlying such effects of Omentin.

Please note that the changes were made as suggested.

2) Para 3.3 Since this paragraph dealing with inflammation is not strictly related to metabolic effect per se’, I propose to join it within paragraph 3.1.

Please note that the changes were made as suggested.

3) 6. Conclusion: “ A schematic diagram summarizing the key features of omentin is presented”. Maybe “effects”, “role” or “actions” should replace the word “features”

Please note that the word “features” was replaced by actions, as suggested.

4) Scheme fig 1: “- Omentin-1 is an adipocytokine widely expressed in a variety of cells, exhibiting microbial defense, and anti-apoptotic properties”. These statement are NOT indicated in figure nor in the main text with dedicated paragraphs. I suggest to omit these actions or discuss them also in the paper (or at least add proper references)

These sentences were removed.

Minor:

1)Para 3.5 “Omentin and adiponectin share structural and functional similarities. Omentin may interact with adiponectin receptors and both adipokines have been linked to improvements in insulin sensitivity [54]. The specific mechanisms through which omentin and adiponectin cooperate in metabolic regulation are still an area of active research” and 4.2.1:” Margaritis and co-workers [70] reported that adipokine expression levels in some adipose tissue depots do not always correlate with adipokine levels in the circulation suggesting intricate mechanisms governing the biology and secretome of adipose tissue [70].” Why are these sentences in Red? Do they represent the only changes made respect to the former original version of the MS? I am not sure. Maybe they should be in black.

Yes, they are now in black.

2) If Fig 1 is taken from a published paper, please add reference in its caption

Its is a diagram that summarizes some of the main actions of omentin-1. It is original.

3) Finally, the way the Conclusions are presented (“- Omentin-1 is an adipocytokine widely expressed in a variety of cells, exhibiting microbial defense, antioxidative, anti-inflammatory, and anti-apoptotic properties.

 - Omentin-1 exhibits a wide range of therapeutic potential in diabetes by reducing comorbidities linked to type 2 diabetes mellitus, such as vascular diseases and diabetic nephropathy [77,78]. Noteworthy, of the major Omentin Energy homeostasis Cardiovascular protection Endothelial function Glucose metabolism Insulin sensitivity Antiinflammation Antioxidant Biomedicines 2023, 11, x FOR PEER REVIEW 8 of 12 adipocytokines that inhibit atherosclerosis, omentin-1 has a strong correlation with inflammation, macrophage differentiation, arterial calcification, and plaque formation [79].

- Omentin-1 inhibits insulin resistance, atherosclerosis, and inflammation through the intracellular signaling pathways of AMP-activated protein kinase/Akt/nuclear factor-κB/mitogen-activated protein kinase (ERK, p38, and JNK).

- Omentin-1 may be used as a biomarker for metabolic syndrome, obesity, diabetes, atherosclerosis, ischemic heart disease, and inflammatory diseases.

- This review sheds light on how omentin-1 might be used to treat these diseases and serve as a biomarker

“) is quite strange to me since they seem a list with each item preceded by a hyphen. Usually, the bullet point is used in a Highlights section. Initially I was thinking this was the caption of Scheme 1. Please, consider revising the presented text to make its role (Caption, highlights or conclusions) clearer.

The conclusion was changed as follows:

“Omentin-1 is an adipocytokine widely expressed in a variety of cells, with antioxidative, and anti-inflammatory properties. This adipokine exhibits a wide range of therapeutic potential in diabetes by reducing comorbidities linked to type 2 diabetes mellitus, such as vascular diseases and diabetic nephropathy [77,78]. In addition, omentin-1 inhibits insulin resistance, atherosclerosis, and inflammation through the intracellular signaling pathways of AMP-activated protein kinase/Akt/nuclear factor-κB/mitogen-activated protein kinase (ERK, p38, and JNK). Noteworthy, of the major adipocytokines that inhibit atherosclerosis, omentin-1 has a strong correlation with inflammation, macrophage differentiation, arterial calcification, and plaque formation [79]. Omentin-1 may be used as a biomarker for metabolic syndrome, obesity, diabetes, atherosclerosis, ischemic heart disease, and inflammatory diseases. This review sheds light on how omentin-1 might be used to treat these diseases and serve as a biomarker.“

 The topic is interesting but it still needs some corrections.
